# Theory and Verification of a new 3D RANS Wake Model

Philip Bradstock[1,2] and Wolfgang Schlez[1]

[1]ProPlanEn Ltd, 71-75 Shelton Street, London, WC2H 9JQ, United Kingdom
[2]Bitbloom Ltd, Desklodge, 1 Temple Way, Bristol, BS2 0BY, United Kingdom

**Correspondence:** Wolfgang Schlez (wolfgang.schlez@proplanen.com)

**Abstract.** This paper details the background to the WakeBlaster model: a purpose built, parabolic three-dimensional RANS solver, developed by ProPlanEn. WakeBlaster is a field model, rather than a single turbine model; it therefore eliminates the need for an empirical wake superposition model. It belongs to a class of very fast (a few core seconds, per flow case) mid-fidelity models, which are designed for industrial application in wind farm design, operation and control.

The domain is a three-dimensional structured grid, a node spacing of a tenth of a rotor diameter, by default. WakeBlaster uses eddy viscosity turbulence closure, which is parameterized by the local shear, time-lagged turbulence development, and stability corrections for ambient shear and turbulence decay. The model prescribes a profile at the end of the near-wake, and the spatial variation of ambient flow, by using output from an external flow model.

## 1 Introduction

In wind farms, wind turbines located downstream of other turbines will experience wake losses. Wind farm development and assessment processes require multiple iterations of configurations, as well as fast project turnaround.

A good understanding of how wake loss works can give a company the competitive edge, while an unexpected systematic performance loss can eliminate the expected profit from a project, or even from an entire project pipeline. Given the importance of wake losses, it may appear contradictory that many in the industry still use analytical single turbine wake models. Using single turbine wake models means that the wake from each turbine is propagated independently, wake expansion is not impacted by neighbouring wakes, and multiple wake deficits are superimposed using an empirical wake superposition model. Single wake models are based on an approach suggested 40 years ago, by Lissaman (1979) and Lissaman et al. (1982), who transferred the work of Abramovich (1963) on free jets to wind turbine wakes. Jensen (1983) presented what is still the most prominent model in this category. Other prominent models of this type include numerical solutions, by Ainslie (1988) and Ott (2011). More recent analytical models include that of Ishihara and Qian (2018).

The longevity of the single wake model approach also speaks for the quality and practical usefulness of these early models. However, in order to provide accuracy for the full range of wind farms (e.g. large wind farms, closely cross-spaced farms, low hub height wind farms, wind farms with stable conditions, or offshore wind farms), an increasing number of empirical corrections had to be made, and parameters added, informed by new experimental data from wind farms, scale experiments, or higher fidelity models - see, for example, Liddell et al. (2005), Schlez et al. (2006), Schlez et al. (2009), and Beaucage et al. (2012). A range of analytical single wake models and superposition methods are reviewed by Porté-Agel et al. (2019).

The increased computational power and scalability available today allows higher fidelity wake models to be used in the iterative process of wind farm design. These models widen the operational envelope, include more physics, and reduce model uncertainties in non-standard situations. The theory behind one such model is presented in this paper: a 3D RANS (Reynolds Averaged Navier-Stokes) wind farm wake model, WakeBlaster.

## 1.1 Related Work

In order to gain a more detailed understanding of wake losses in a wind energy research context, two groups of 3D-RANS codes have been developed. The models are referred to as 'field models', to distinguish them from the single turbine models by Crespo et al. (1999).

The first group of 3D-RANS codes are parabolic solvers, using the thin shear layer approximation, see Ferziger et al. (1997). Parabolic solvers assume a dominant flow direction and information is transported only downstream. Crespo et al. developed UPMWAKE at UPM (Universidad Polytechnica de Madrid), and later Crespo et al. (1994) developed an extension for wind farms, called UPMPARK. A number of further variants have been developed and reviewed by Vermeer et al. (2003). One branch was continued by TNO/ECN (The Energy Research Center of the Netherlands), and it resulted in the WakeFarm presented by Schepers (2003), and FarmFlow model presented in Eecen et al. (2011). Renewed interest in mid-fidelity models has recently led to the independent development of several new models in this group, like those presented by Trabucchi et al. (2017) and Martinez-Tossas (2019).

The second group of 3D-RANS field models, the elliptic solvers, is more widespread. Elliptic solvers are generally more powerful, and they iterate equations numerically, in order to allow information to be transported in all directions; this makes them more expensive computationally (by several orders of magnitude). These models use a k-$\epsilon$ or k-$\omega$ turbulence closure, describing the generation and dissipation of turbulent kinetic energy. Models in this group are (in principle) also capable of solving upstream effects, such as the interaction of wakes in the induction zone, and the near wake of wind turbines. Some models are based on general purpose flow solvers, whereas others are in-house developments - examples can be found in publications by Crespo et al. (1988); Prospathopoulos et al. (2010); Barthelmie et al. (2011); van der Laan et al. (2017); Michelsen (1994).

The WakeBlaster model developed by ProPlanEn by Schlez et al. (2017b) belongs to the parabolic solver group. A parabolic solution offers a good balance between improved accuracy, additional detail, and computational costs. The target of the new model is to improve the accuracy of wind farm loss modelling. Two specific aims are to address the interaction between wakes, as well as the interaction between wakes and the atmospheric boundary layer for different levels of atmospheric stability. Special attention was paid to the validation of the model, using data from a wide range of wind farms and atmospheric conditions, which has been reported by the authors in Schlez et al. (2017a, 2018, 2019); Bradstock et al. (2018); Braunheim et al. (2018), and independently evaluated and compared to engineering models in a blind test for offshore wind farms, by Sanz et al. (2019).

The fundamental equations and assumptions for this solver are shown in the following Section 2. Section 3 presents as example the verification of the model for an offshore wind farm and the results of verifying the computational performance. Section 4 discusses model limitations, followed by the conclusions in Section 5.

## 2 Theoretical Background

The WakeBlaster wind farm simulator is based on a Reynolds-Averaged Navier-Stokes (RANS) set of equations, which is used to solve the propagation of wake dissipation through the farm domain, in Cartesian 3D coordinates. In order to account for the fluctuation term of the velocity vector, it uses eddy viscosity turbulence closure, where the eddy viscosity is calculated from the combined wake and ambient wind speed shear profiles.

### 2.1 RANS Equations

The wake model uses RANS equations for momentum conservation, and mass flow conservation to calculate the three components of wind velocity in the axial, lateral and vertical directions. Cartesian 3D vectors are used for displacement $\vec{\mathbf{x}}$ and wind speed relative to ambient $\vec{\mathbf{u}}$: $\vec{\mathbf{x}} = \begin{bmatrix} x, y, z \end{bmatrix}$ $\vec{\mathbf{u}} = \begin{bmatrix} u, v, w \end{bmatrix}$, where the first element of the vectors $(x)$ is the streamwise component, the second element $(y)$ is horizontal and perpendicular to $(x)$, and the third element $(z)$ is vertical (starting from the ground up) and makes up a right-hand coordinate system.

The Reynolds averaged momentum and mass conservation equation can be expressed in two dimensions, for either a free jet or a wake submerged in an incompressible fluid, as given by Abramovich (1963):

$$\frac{\partial u}{\partial t} + \frac{\partial u^2}{\partial x} + \frac{\partial \overline{u'u'}}{\partial x} + \frac{\partial (uv)}{\partial y} + \frac{\partial \overline{u'v'}}{\partial y} = \nu \frac{\partial^2 u}{\partial y^2} - \frac{1}{\rho} \frac{\partial p}{\partial x} \tag{1}$$

representing the momentum in the flow direction, where $u'$, $v'$ and $w'$ denote fluctuations from mean values, and $\nu$ the viscosity and $\rho$ the density of the fluid. The corresponding continuity equation is

$$\frac{\partial u}{\partial x} + \frac{\partial v}{\partial y} = 0 \tag{2}$$

The momentum equations in transversal directions are not considered in the description of a free jet or wake present beyond the near wake of a wind turbine.

### 2.2 Simplifying Assumptions

The following simplifying assumptions are applied by Abramovich for a stationary free wake, expanding into an infinite region:

**Viscosity** The effect of molecular viscosity is small $\nu \frac{\partial^2 u}{\partial y^2} = 0$ compared to the turbulent viscosity

**Pressure** Flow pressure gradients can be neglected in most cases $\frac{1}{\rho} \frac{\partial p}{\partial x} = 0$

**Stationary** The flow is stationary with respect to the mean velocities $\frac{\partial u}{\partial t} = 0$

**Thin shear layer approximation** Fluctuations along the flow change much slower than in the transversal direction $\frac{\partial \overline{u'u'}}{\partial x} = 0$

After substituting the continuity equation and applying the simplifying assumptions Abramovich (1963) obtains:

$$u \frac{\partial u}{\partial x} + v \frac{\partial u}{\partial y} + \frac{\partial \overline{u'v'}}{\partial y} = 0 \tag{3}$$

or expanded to three dimensions:

$$u\frac{\partial u}{\partial x} + v\frac{\partial u}{\partial y} + w\frac{\partial u}{\partial z} + \frac{\partial \overline{u'v'}}{\partial y} + \frac{\partial \overline{u'w'}}{\partial z} = 0 \tag{4}$$

Using the Boussinesq eddy viscosity assumption, the stress components $\overline{u'v'}$ and $\overline{u'w'}$ are expressed as:

$$\overline{u'v'} = -\epsilon\left(\frac{\partial u}{\partial y} + \frac{\partial v}{\partial x}\right) \approx -\epsilon\frac{\partial u}{\partial y} \qquad \overline{u'w'} = -\epsilon\left(\frac{\partial u}{\partial z} + \frac{\partial w}{\partial x}\right) \approx -\epsilon\frac{\partial u}{\partial z} \tag{5}$$

where $\epsilon$ denotes the isotropic eddy viscosity. The streamwise variation in transversal velocities ($\frac{\partial v}{\partial x}$ and $\frac{\partial w}{\partial x}$) is small compared to the transversal variation of streamwise velocity ($\frac{\partial u}{\partial y}$ and $\frac{\partial u}{\partial z}$). The spatial variation in eddy viscosity can be neglected in first approximation and is therefore approximated as a constant, the governing momentum conservation equation can now be written as:

$$u\frac{\partial u}{\partial x} + v\frac{\partial u}{\partial y} + w\frac{\partial u}{\partial z} - \epsilon\frac{\partial^2 u}{\partial y^2} - \epsilon\frac{\partial^2 u}{\partial z^2} = 0 \tag{6}$$

### 2.3 Numerical Solution

The ambient wind field is determined by an external flow model, and it determines the inflow conditions and spatial variations over a site. The turbine is represented by its hub height, diameter and other readily available and measured characteristics.

#### 2.3.1 Model Domain

The waked wind field is set up by creating a two-dimensional flow plane, which forms a cross-section along the $y$ and $z$ axes of the velocity vector $\overline{u}$. The flow plane is propagated downstream along the $x$ coordinate and when it passes a turbine, a wake is injected into the flow plane.

The grid spacing is set by default to a tenth of rotor diameter. In the vertical direction, the grid starts at the ground $z = 0$ and reaches up to a default height of three rotor diameters or 31 grid layers. In the horizontal direction, the grid is expanded, as required, to enclose each wake injected into the flow plane with an additional four rotor diameters to the side to allow for wake expansion.

#### 2.3.2 Wind Turbine Momentum Extraction

Axial-momentum theory prescribes pressure building up in the induction zone upstream of any wind turbine or wind farm, and pressure recovery in the near-wake downstream of the rotor. The momentum that each of the turbines extracts in the process is the wind speed dependent thrust coefficient, as a function of the idealised incident wind speed, $U_{inc}$, at each turbine location, without the presence of the turbine.

In the model, the momentum deficit is injected at the end of the near-wake (which is assumed to be at 2 diameters downstream of the rotor) of each turbine, and it is distributed over an expanded rotor area, using the blunt bell-shaped wind speed deficit profile from Lissaman et al. (1982). The centre-line wind deficit relative to incident wind speed $D_m$, experimentally determined by Ainslie (1988) at a downstream distance of 2 diameters is used as a function of inflow turbulence $I_{inc}$ and thrust coefficient $c_t$.

$$D_m = c_t - 0.05 - (16c_t - 0.5)\frac{I_{inc}}{10} \tag{7}$$

The radial width of the profile is then derived by ensuring momentum conservation with regard to the thrust coefficient of the turbine.

### 2.3.3 Flow Plane Propagation

The flow plane is propagated according to equation 6 using the alternating direction implicit (ADI) method described by Peaceman and Rachford (1955); von Rosenberg (1983), where it is alternately solved in the $xy$ and $xz$ planes, incrementing the $x$ (downstream) coordinate by half a propagation step between each solving plane, so that both planes are solved once per step. By solving for each row or column in the flow plane, and by employing the central difference method, the problem can be set up numerically in a tridiagonal matrix equation, which can then be solved efficiently for the axial velocity, $u$, by the Thomas algorithm Thomas (1949), described for example in Burden and Faires (2001). In 3D Cartesian coordinates the tridiagonal equation must be solved for every row or column of the flow plane, depending on which direction a solution is obtained. Dirichlet boundary conditions are used by enforcing $u = 1$ in the extremities of the flow plane.

At each half-step of the solving process, the horizontal and vertical velocities, $v$ and $w$ respectively, are calculated for all points in the flow plane according to 2. For any given step there are two unknowns in this equation, $v$ and $w$, and therefore it cannot be solved analytically in a single step. Instead, the unknowns are calculated numerically, by calculating each individually, and iterating until their values converge. By rearranging equation 2, $v$ and $w$ can be expressed individually for a parabolic flow:

$$v = -\int \frac{\partial u}{\partial x} + \frac{\partial w}{\partial z} dy \quad ; \quad w = -\int \frac{\partial u}{\partial x} + \frac{\partial v}{\partial y} dz \tag{8}$$

In practice, due to the assumption of incompressibility, this formulation will lead to a local velocity shear, resulting in non-zero lateral and vertical velocities that are infinitely far from the source of shear. In reality this would not be the case, due to the compressibility of air. Therefore, in order to account for the effect of compressibility, a spatial damping term is introduced, so that $v$ and $w$ tend to zero at $y = -\infty$, $y = \infty$ and $z = \infty$:

$$v = -\int \left( \frac{\partial u}{\partial x} + \frac{\partial w}{\partial z} - \gamma v \right) dy \quad ; \quad w = -\int \left( \frac{\partial u}{\partial x} + \frac{\partial v}{\partial y} - \gamma w \right) dz \tag{9}$$

where $\gamma$ is a user-configurable positive constant that determines the strength of lateral and vertical velocity damping. As these integrals are indefinite, boundary conditions must be assigned. In the vertical direction, it is given that vertical velocity at

ground level is zero, as mass flow cannot pass into or out of the ground. Therefore, the condition $w_{z=0} = 0$ is applied, leading to:

$$w(z) = -\int_0^z \left( \frac{\partial u}{\partial x} + \frac{\partial v}{\partial y} - \gamma w \right) dz' \tag{10}$$

In the lateral direction, the physical boundary conditions are that $v_{y=-\infty} = v_{y=\infty} = 0$, because the wind farm wakes cannot induce lateral velocity far from the farm. However, for numerical purposes, the size of the flow plane is constrained, and it cannot be guaranteed that the velocity will reach zero on both sides of the flow plane. Therefore, the lateral velocity is integrated in each direction, starting from zero, and the mean of the two is taken. This is expressed as:

$$v(y) = -\frac{1}{2} \int_{y_{min}}^y \left( \frac{\partial u}{\partial x} + \frac{\partial w}{\partial z} - \gamma v \right) dy' + \frac{1}{2} \int_{y_{max}}^y \left( \frac{\partial u}{\partial x} + \frac{\partial w}{\partial z} - \gamma v \right) dy' \tag{11}$$

where $y_{min}$ and $y_{max}$ are the lateral location of the edge of the flow plane.

## 2.4 Eddy Viscosity Calculation

The key term controlling the rate of wake dissipation is eddy viscosity. Eddy viscosity has dimensions of length squared over time, and it can be represented by multiplying a length scale of the shear layer by a velocity scale of the flow field.

WakeBlaster calculates eddy viscosity from the shear profile of axial velocity in the $yz$ plane. In order to do this, it creates a combined flow plane of the ambient wind speed, $U_{amb}$, multiplied by the solved wake flow plane, $u$, which is relative to ambient wind speed. In neutral atmospheric conditions, the ambient wind speed is calculated as a logarithmic function of height above ground:

$$U_{amb}(z) = \frac{u^*}{\kappa} \ln \frac{z}{z_0} \tag{12}$$

where $u^*$ is the friction velocity, taken to be 2.5 times the value of standard deviation of the axial wind velocity, $\kappa$ is the von-Karman constant (value = 0.4) and $z_0$ is the roughness length. The unknown parameters are determined from inputs to the simulation, such as wind speed and turbulence intensity at a particular height (usually the hub height of one of the turbines). The eddy viscosity is then calculated for every point in the flow plane, using the following process:

1. Create a combined flow plane by multiplying the ambient surface layer wind speed profile by the waked flow plane
velocity $u$.

2. For each point, identify the local minimum and maximum velocity. For a point located at $(y, z)$, local is determined as the range $[y - \eta z, y + \eta z]$ and $[(1-\eta)z, (1+\eta)z]$, in the lateral and vertical directions respectively, where $\eta$ is a configurable constant which meets the criterion $0 < \eta < 1$.

3. In each of the two directions, the component of eddy viscosity is calculated as $\epsilon_i = \Delta u_i \Lambda_i$, where $\Delta u_i$ is the difference between minimum and maximum velocity and $\Lambda_i$ is the distance between the maximum and minimum points. This process is shown in figure 1.

4. The overall eddy viscosity is the calculated as $\bar{\epsilon} = k\sqrt{\epsilon_y^2 + \epsilon_z^2}$, where $k$ is a positive calibration constant, which although configurable is considered to be independent of wind farm size and layout.

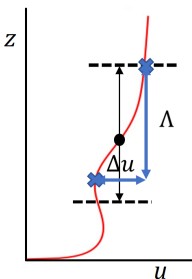

**Figure 1.** Calculation of the vertical component of eddy viscosity by finding the points of minimum and maximum velocity within a given height range.

For a logarithmic wind speed profile in the vertical direction with no lateral variation, this method leads to an eddy viscosity that is proportional to the height above ground.

### 2.4.1 Eddy Viscosity Lag

The eddy viscosity, as so far described in section 2.4, is solely based on the wind shear profile. However, no newly created shear profile instantly generates turbulence, and therefore eddy viscosity - in reality, there is a lag between the change in a shear profile and its effect upon eddy viscosity and wake dissipation. In WakeBlaster this lag is formulated in terms of downstream distance, and it has two distinct models.

The 'fixed' model obeys a first order lag equation:

$$\ell \Lambda \frac{d\epsilon}{dx} + \epsilon = \bar{\epsilon} \tag{13}$$

where $\epsilon$ is the lagged eddy viscosity, $\Lambda$ is the length-scale defined in the previous section, and $\ell$ a positive constant defining the lag length relative to the length scale and is considered to be independent of wind farm size or layout.

The 'turbulence dependent' model gives a larger lag distance when the eddy viscosity and turbulence are low, and it obeys the following equation:

$$\frac{\Lambda}{\phi \frac{\epsilon}{kz} + \frac{\Lambda}{\lambda_{max}}} \frac{d\epsilon}{dx} + \epsilon = \bar{\epsilon} \tag{14}$$

where $\phi$ is a positive parameter that determines the strength of turbulence on the lag length, and $\lambda_{max}$ is also a positive parameter that corresponds to the lag length when turbulence is zero. Both parameters are calibrated against extensive wind farm observational data and are considered to be independent of wind farm size and layout.

### 2.4.2 Atmospheric Stability

When simulating atmospheric conditions that are not neutral, the calculation of eddy viscosity is modified. This modification uses the Monin-Obukhov length, $L$, and the concept of non-dimensional wind shear, $\phi_m$, which is defined by Businger (1971), as:

$$\phi_m = \frac{\kappa z}{u*} \frac{\partial U}{\partial z} \tag{15}$$

Furthermore, according to Businger (1966), the non-dimensional wind shear is empirically approximated as what tends to be known as the Businger-Dyer relationship:

$$\phi_m = \begin{cases} 1 + 5\zeta & \text{stable } (L > 0) \\ 1 & \text{neutral } (L \text{ undefined}) \\ (1 - 16\zeta)^{-\frac{1}{4}} & \text{unstable } (L < 0) \end{cases} \tag{16}$$

where $\zeta = \frac{z}{L}$. The ambient wind speed shear profile is then modified by introducing $\psi_m$:

$$U_{amb}(z) = \frac{u*}{\kappa} ln\left( \frac{z}{z_0} + \psi_m(\zeta) \right) \tag{17}$$

where:

$$\psi_m = \int_{\zeta_0}^{\zeta} [1 - \phi_m] d\zeta \tag{18}$$

where $\zeta_0 = \frac{z_0}{L}$. Furthermore, the vertical component of the eddy viscosity, $\epsilon_z$, is also modified by the non-dimensional wind shear:

$$\epsilon_z = \frac{\Delta u_z \Lambda_z}{\phi_m} \tag{19}$$

The horizontal component of eddy viscosity $\epsilon_y$, is left unmodified.

## 2.5 Wind Turbine Power Calculation

WakeBlaster calculates the power output using power curve input from the user. In order to calculate accurate power, corresponding to the variant wind speed across the rotor, a rotor equivalent wind speed ($U_{rot}$) is calculated. This is done by first calculating the combined ambient and wake axial velocity ($U = U_{amb}u$ at the rotor plane), and then integrating across the rotor disk area:

$$U_{rot} = \sqrt[n]{\int_A U^n dA} \tag{20}$$

where $n$ is an integer. A popular approach is to use $n = 3$ as suggested in IEC61400-12-1 (2017), based on the principle that the power available in the wind is proportional to the cube of the wind speed. However, WakeBlaster uses a value of $n = 1$ by default as turbines will not be able to realise the full potential of a sheared inflow over the rotor. As this method is performed on the combined ambient and wake axial velocity, the effects of wind shear on power production are implicitly included whenever the severity of the wind shear depends on the turbulence and atmospheric stability of the flow case.

A general directional variability of the wind within each flow case is included in a standard power curve. A rotor yaw angle can be set per turbine, to consider in the power calculation a known average directional misalignment with the rotor plane. A model to modify the power curve for site specific directional variability over the rotor, for example changes with height or for specific meteorological conditions, is not included in the model.

WakeBlaster uses IEC methods in IEC61400-12-1 (2017) to adjust the power curve for air density and turbulence intensity. The rotor equivalent turbulence intensity is also calculated using the integral method above, but instead using a value of $n = 2$.

## 3 Verification

In this section, the grid dependence and sensitivity is analysed and an estimate of the numerical uncertainty is thereby provided. Computational performance for large wind farms is verified, and offshore wind farm model predictions are inspected graphically, for plausibility.

### 3.1 Grid Dependence and Sensitivity

The model uses a structured grid, in terrain following coordinates. The grid resolution is scaled with a length scale characterising the specific flow - the rotor diameter. The grid is equally spaced in all directions, and no stretching, compression, or nesting is applied to any part of the domain. The minimalist design is computationally efficient, and it avoids potential numerical errors - at grid interfaces which do not match, for example.

The solver is designed for a single purpose: to model the impact of wind turbines on the underlying flow and the consequential wind farm wake losses. A wind turbine's wake scales with its rotor diameter and its height above ground. In order to match

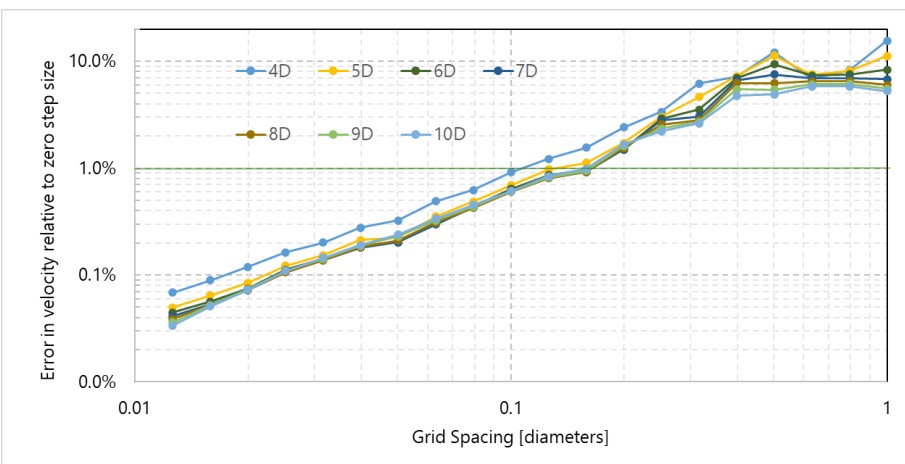

**Figure 2.** Numerical error due to the grid spacing, based on the difference in wind speed from the hypothetical zero grid spacing value calculated by Richardson extrapolation. The scenario assessed has an ambient wind speed of 8m/s and $c_t = 0.8$, neutral atmospheric stability and 10% turbulence. The acceptable numerical error is shown as 1%.

the dominant scale in the flow for each wind farm, the grid resolution is fixed at 0.1 diameters; it thus scales with the rotor diameter.

Analysis of the sensitivity of model results to changes in grid resolution verifies that the results are not sensitive to grid resolution over the expected range of application. Challenges could arise - for example, when using an average resolution in wind farms with mixed turbine diameters and turbines mounted at low hub heights. In an annual energy calculation, the overall

wake loss is composed of several thousand individual flow cases. Wake loss model errors are commonly estimated to be in the range of 10-20 percent, relative to the average annual wake loss. Numerical errors should be one order of magnitude lower. Ignoring error compensation between flow cases, an error of 1-2 percent (relative to the wind speed difference for an individual flow case) is acceptable.

The grid dependency study was carried out for the following scenario: A single turbine, with ambient wind speed perpen-

245 dicular to the rotor plane. Ambient conditions were a wind speed of 8 m/s, neutral atmospheric stability, and a turbulence intensity of 10%. The wind turbine type (V100-1.8) is described by its geometry and thrust coefficient, which is $c_t = 0.8$ for the scenario. The key target value investigated is the wind speed relative to ambient wind speed, at hub height and at several distances downstream.

The sensitivity was tested in a flow case with a strong wake, and the results are presented in 2. The error in wind speed is

250 presented relative to a hypothetical error value, which is calculated using a Richardson extrapolation for an infinitesimal grid spacing, as suggested by Roy (2003) for mixed order numerical schemes.

The numerical error, due to grid spacing for an operational range of up to just above 0.1 diameters, is below 1 %. At a coarser resolution the model can no longer resolve the structure of the flow sufficiently. The current choice of grid resolution (0.1 diameters) represents a reasonable compromise between computational efficiency and model accuracy.

The grid resolution in the model scales automatically with the rotor diameter. Neither the grid nor the resolution are variables which should (under normal circumstances) be adjusted by any user.

## 3.2    Computational Performance

WakeBlaster is a medium-fidelity tool, which is typically capable of running each flow case in a few seconds, on the single core of a modern processor. With the default settings (a flow plane resolution of 0.1 rotor diameters and a domain height of three
diameters), the time (in seconds) to run a single flow case ($T_{fc}$) is (on an Intel i5 8[th] generation processor) approximately:

$$T_{fc} \approx 0.0015 \frac{A}{D^2} \text{ s} \tag{21}$$

where $A$ is the area of the wind farm and $D$ is the rotor diameter. The $T_{fc}$ is proportional to the area of the wind farm, and (at equal turbine density) to the number of wind turbines in the wind farm. However, the exact time will depend on the wind farm's layout, the wind direction, and the architecture of the processor. The $T_{fc}$ is also proportional to the cube of the flow
plane resolution, although results do not show any significant improvement in accuracy when the resolution is increased.

     For example, a typical flow case for Horns Rev - a wind farm with 80 turbines arranged in a grid, with inter-turbine spacing of 7 diameters - runs in about 5 s. Unless hysteresis effects are included in a time series simulation, each required flow case remains independent from the others, allowing many flow cases to be run in parallel. As WakeBlaster is hosted on the cloud, this allows a high level of parallelisation across tens or hundreds of processors, meaning that an energy assessment consisting
of (for example) 2,000 flow cases can be completed in a matter of minutes, even for large wind farms.

## 3.3    Visual Verification

Using a three-dimensional wake model, it is possible to create plots of the three-dimensional waked flow field for the complete wind farm, for a particular flow case. This article presents a visualisation of a single flow case from the Lillgrund wind farm, located in the Øresund Strait, between Sweden and Denmark. The Lillgrund wind farm presents a good case study, because
the small spacing between turbines (3.3 and 4.3 rotor diameters, along the two principal rows) leads to large wake effects. The layout is shown in figure 3. The turbines have a rotor diameter of 93 m and a hub height of 68 m above mean sea level.

     Three cross-sectional slices in the $xy$, $xz$ and $yz$ planes, for a flow case of 8 m/s wind speed, 270 deg wind direction, 6 % turbulence intensity, and neutral atmospheric conditions, are presented in figure 4.

     These simulations indicate that there is significant interaction between wakes originating from individual turbines, and this
supports the assumption that the wakes cannot be modelled independently. The wake interaction leads to a complex wake shape downstream of the wind farm. The low hub height of the wind turbines (68 m), relative to their rotor diameter (93 m), results in significant ground-wake interaction effects. As ambient mixing from below is limited, single turbine wakes become asymmetrical in shape, and the point of greatest deficit drifts downwards to below hub height.

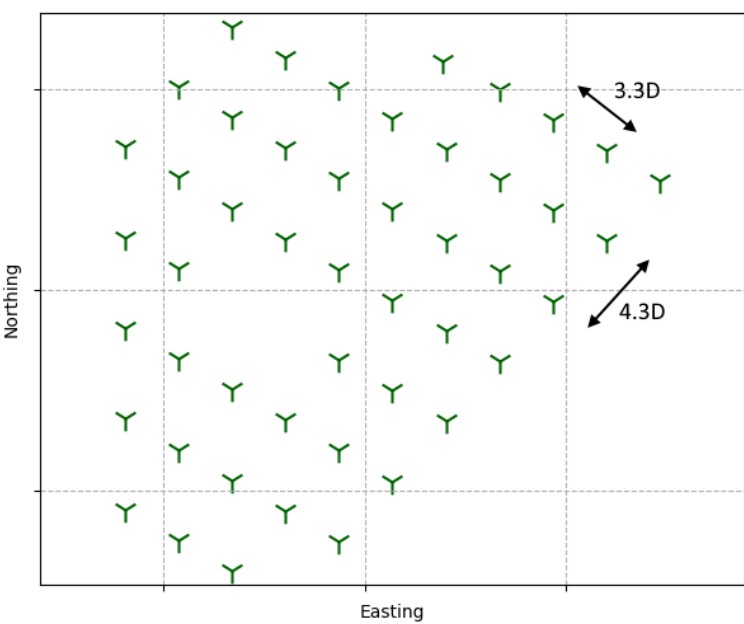

**Figure 3.** Layout of the Lillgrund wind farm. The turbine rotor diameter is 93 m with a hub height of 68 m. The turbine spacing is approximately 4.3 rotor diameters, along the South-West to North-East rows, and 3.3 diameters along the South-East to North-West rows.

## 4 Limitations

The code is a mid-fidelity code designed to be fast and capable of simulating projects with several thousand turbines, working with limited amount of readily available input data and be used in an iterative design process. This limits the level of detail that can be included in the sub-models.

– No direct interaction between the turbines and no description of the axial pressure gradient are included in the model. The induction zones directly upstream and downstream (near wake) of turbines can overlap and interact. This may lead to changes in turbine performance and turbine characteristics and no attempt has been made to quantify such effects.

– A basic representation of the the ambient flow is used as input to the model. The wake is modelled as a perturbation of the underlying flow. No attempt has been made to model a two-way interaction with the atmospheric boundary layer.

– The model uses the directional speedups predicted by a suitable flow model (for example in a RSF/WRG format) to account for spatial variation of the wind resource, for example due to orography, or roughness. Further complex terrain effects, like flow separation, are not considered.

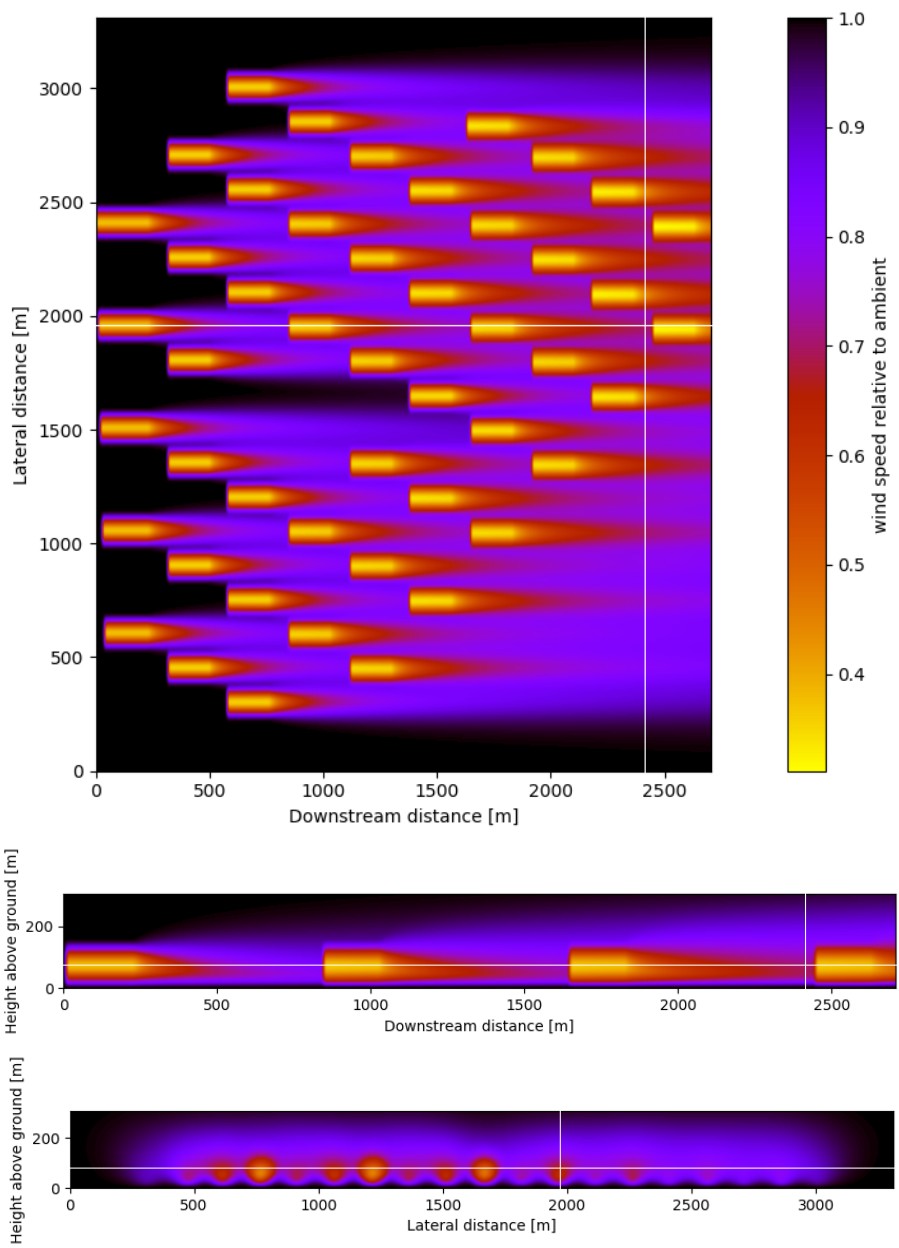

**Figure 4.** Plots of the axial velocity in the wind farm relative to ambient wind speed for a flow case of 8 m/s with the wind from due West. From top to bottom: $xy$ (birds-eye) slice at hub height; $xz$ (side-on); $yz$ (front-on). The white lines show the corresponding planes of the other plots. The $xy$ plot is taken at the turbine hub-height above sea-level - 68 m.

- The ambient wind direction is assumed to be constant throughout the wind farm. Therefore in curved flows (due to terrain or due to meteorological factors), downstream wake locations may not be accurate.

The WakeBlaster model undergoes continuous, data driven improvement, and refined models will be added successively.

## 5  Conclusions

This is the first publication to present the theoretical background of WakeBlaster in some detail. WakeBlaster is a recently developed 3D-RANS solver that is specialised to simulate the waked flow field in wind farms. The characteristics of this model show the desired performance balance between speed and level of detail.

*Code availability.*  WakeBlaster calculations are provided as a cloud service and designed for integration in other software packages. Wake-Blaster is available from ProPlanEn directly (www.wakeblaster.net) and through third party implementations. WakeBlaster has been integrated in EMD's WindPro software and is scheduled for release in May 2020.

*Author contributions.*  Philip Bradstock: formal analysis, investigation, methodology, software development, data curation, verification, visualisation, writing; Wolfgang Schlez: conceptualisation, funding acquisition, project administration, resources, investigation, supervision, methodology, writing

*Competing interests.*  WakeBlaster is a commercial product of ProPlanEn Ltd. Wolfgang Schlez is the founder and sole share holder of ProPlanEn Ltd. Philip Bradstock was employed by ProPlanEn Ltd at the time of carrying out the model development. He is a director of Bitbloom Ltd. providing services to ProPlanEn Ltd.

*Acknowledgements.*  The development of WakeBlaster was co-funded by the UK's innovation agency, Innovate UK. A number of companies contributed operational wind farm data to this research, and their support is greatly appreciated. ProPlanEn GmbH processed the Lillgrund test case, as part of IEA task-31: WakeBench. The contributions of Staffan Lindahl and Sascha Schmidt (verification and visualisation), Michael Tinning (software development) and Vassilis Kostopoulos (verification,investigation, and methodology) are acknowledged.

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
