# Peer review of "Theory and Verification of a new 3D RANS Wake Model"

_Wind Energy Science, 2020_

## Referee Comment (RC1) · Paul van der Laan (Referee) · 26 Mar 2020

**Review of *Theory and Verification of a new 3D RANS Wake Model* by Philip Bradstock and Wolfgang Schlez**

Reviewer: M. Paul van der Laan, DTU Wind Energy

The authors present a parabolic Reynolds-averaged Navier-Stokes (RANS) model for 3D wind farm flows. The assumptions and limitations are discussed and an example wind farm simulation is presented. I appreciate the effort of making a scientific publication of a commercial wake model. However, the current content of the article is more suited as a technical report. In addition, the title is somewhat misleading because the model is not verified. A model verification of a RANS model would involve a grid refinement study. I also miss a validation of the model. If both a proper verification and validation are added then it has the potential to be published as scientific article in Wind Energy Science. Therefore, I recommend a major revision. In addition, I have listed major and minor comments below that need to be addressed:

**Main comments**

1. In the abstract you mention: *The WakeBlaster model is verified, calibrated and validated using a large volume of data from multiple onshore and offshore 10 wind farms.* I cannot find a reference to this work and it is also not included in the present work. I would recommend to include the calibration, verification (grid refinement study) and validation in this work. Otherwise, the scientific content is not sufficient for a publication in Wind Energy Science. In addition, please note that the abstract should include a motivation, a short summary of the work, and the main results and conclusions, so it cannot contain conclusions based on previous work.

2. How are the results post processed when the annual energy production is evaluated? For example, do you include a Gaussian filter to represent wind direction uncertainty? (As introduced by Gaumond et al. (2014) and applied for RANS in van der Laan et al. (2015a)).

3. Since the authors are both employees at ProPlanEn, a commercial entity that is selling the presented model, it would make sense to mention this in the Section Competing interests.

4. Can WakeBlaster handle (complex) terrain? If not, I would mention that the model can only used for wind farms in flat terrain and offshore conditions.

5. Pages 2-3, Lines 58-60: I do not understand what you mean by *In order to account for the unsteady terms, it uses eddy viscosity turbulence closure, where the eddy viscosity is calculated from the combined wake and ambient wind speed shear profiles.* I guess you mean turbulent fluctuations instead of unsteady terms. Unsteady terms can be handled by including a transient term, as can be done in Unsteady RANS (URANS).

6. The reference to Abramovich (1963) is not very accessible. In addition, eq. (2) of the article is the boundary layer equation without viscous effects and I do not understand how this equation can be extended to three dimensions because the original boundary layer equation describes a streamwise $U$ and vertical velocity, which in your coordinate system is $W$ not $V$. I would suggest to start Section 2.1 with the 3D RANS equations including external forces (e.g. wind turbine

thrust force) but without viscous effects:

$$\frac{\partial U_i}{\partial x_i} = 0, \tag{1}$$

$$U_j \frac{\partial U_i}{\partial x_j} = f_i - \frac{1}{\rho}\frac{\partial P}{\partial x_i} + \frac{\partial \overline{u_i' u_j'}}{\partial x_j}$$

where $x_i$ are the Cartesian coordinates, $U_i$ is the mean velocity vector, $P$ is the mean pressure, $f_i$ are the external forces and $\overline{u_i' u_j'}$ are the Reynolds-stresses. The equations in full form can be written as:

$$\frac{\partial U}{\partial x} + \frac{\partial V}{\partial y} + \frac{\partial W}{\partial z} = 0, \tag{2}$$

$$U\frac{\partial U}{\partial x} + V\frac{\partial U}{\partial y} + W\frac{\partial U}{\partial y} = f_1 - \frac{1}{\rho}\frac{\partial P}{\partial x} + \frac{\partial \overline{u'u'}}{\partial x} + \frac{\partial \overline{u'v'}}{\partial y} + \frac{\partial \overline{u'w'}}{\partial z},$$

$$U\frac{\partial V}{\partial x} + V\frac{\partial V}{\partial y} + W\frac{\partial V}{\partial z} = f_2 - \frac{1}{\rho}\frac{\partial P}{\partial y} + \frac{\partial \overline{u'v'}}{\partial x} + \frac{\partial \overline{v'v'}}{\partial y} + \frac{\partial \overline{v'w'}}{\partial z},$$

$$U\frac{\partial W}{\partial x} + V\frac{\partial W}{\partial y} + W\frac{\partial W}{\partial z} = f_3 - \frac{1}{\rho}\frac{\partial P}{\partial z} + \frac{\partial \overline{u'w'}}{\partial x} + \frac{\partial \overline{v'w'}}{\partial y} + \frac{\partial \overline{w'w'}}{\partial z}$$

In order to arrive at eq. (3) of the article, the following additional assumptions are necessary:

(a) The momentum equations for $V$ and $W$ are ignored.

(b) The streamwise pressure gradient is zero: $\frac{\partial P}{\partial x} = 0$.

(c) The streamwise external force is zero: $f_1 = 0$.

(d) The gradient of the normal Reynolds-stress in the streamwise direction is zero: $\frac{\partial \overline{u'u'}}{\partial x} = 0$.

and then we get eq. (3) of the article:

$$U\frac{\partial U}{\partial x} + V\frac{\partial U}{\partial y} + W\frac{\partial U}{\partial y} = \frac{\partial \overline{u'v'}}{\partial y} + \frac{\partial \overline{u'w'}}{\partial z} \tag{3}$$

Applying the Boussinesq approximation for the remaining Reynolds-stresses we obtain:

$$\overline{u'v'} = \nu_T \left( \frac{\partial U}{\partial y} + \frac{\partial V}{\partial x} \right), \qquad \overline{u'w'} = \nu_T \left( \frac{\partial U}{\partial z} + \frac{\partial W}{\partial x} \right) \tag{4}$$

In order to arrive at eq. (5) of the article we need to assume that $\partial V/\partial x$ and $\partial W/\partial x$ are zero, which is not mentioned in the article. The assumptions (a) and (c) are neither mentioned. In addition, if you assume that the eddy-viscosity is a constant, then you are basically solving for a laminar flow because one could replace the eddy-viscosity with a molecular viscosity. In other words, one could ignore the Reynolds-stresses and include the viscous terms in the $U$-momentum equation in order to arrive at the same equation. The work could be approved a lot if you can quantify the errors made by each assumption or quantify the contribution to the wake of each ignored term, which could be based on an elliptic RANS model, as performed by Iungo et al. (2018).

7. Section 2.3.1: You mention that the near wake stream-wise velocity profile is prescribed for each wind turbine. How do you determine the end of the near wake and how does it vary with the wind turbine thrust coefficient and atmospheric conditions as turbulence intensity and stability? Do I understand correctly that eq. (7) is used to determine the initial magnitude of the centerline wake deficit at a defined downstream location? In addition, eq. (7) is derived from wind tunnel measurements where the turbulence length scales and Reynolds-number are very different from utility scale wind

turbines, so would that mean eq. (7) needs to be recalibrated? Finally, I was wondering why you are not modeling the wind turbine thrust force as an external force in the streamwise momentum equation instead of prescribing a velocity deficit profile in the near wake.

8. Section 2.4: The eddy-viscosity is no longer a constant, as assumed in eq. (5) of the article, which is inconsistent. Please motivate and clarify.

9. There are a number of undefined parameters and constants. What are the values of $k$ (Page 6, line 158), $\ell$ (eq. 13), $\phi$ (eq. 14), $\lambda_{\max}$ (eq. 14)?

10. Should $k$ be $\kappa$ in eq. (14)?

11. Section 2.4.1. I suspect that the eddy-viscosity lag model could be replaced or simplified using a length scale limiter in the form of an $f_P$ function ($\nu_T^* = \nu_T f_P$) that only has one constant to calibrate, see for example van der Laan et al. (2015b) or van der Laan and Andersen (2018).

12. Section 3.1: Please motivate the chosen grid resolution of $D/10$, where $D$ represent the rotor diameter, using a grid refinement study. In addition, a domain height of $3D$ seems very low to me, please show that this domain height does not affect the solution. I normally use $25D$ for 3D elliptic RANS simulations of wind farms. What are the other dimensions of the 3D flow domain? Do you use stretching of cells in order to reduce the total number of cells or is the domain discretized uniformly?

13. Section 3.1: You mention that a flow case of Horns Rev I takes 5 s. (Please briefly introduce the Horns Rev I wind farm here). That would mean an annual energy production calculation of 22 wind speeds and 360 wind directions would take 11 hours on a single CPU. This can be made parallel as you mention (using a few hundred cores). However, WakeBlaster should provide more accurate results compared to an engineering wake model that can calculate the AEP in about 1 s on a single core in order to make sense to run. Therefore, I would suggest to both validate WakeBlaster with wind farm measurements and compare the performance with one or two engineering wake models in the present work.

14. Section 3.2: You could use this wind farm as both a verification (grid study) and validation case. Presenting a show case is not enough for a scientific document.

15. Section 4: You could add that only flat terrain is considered. In addition, I do not agree that meandering of the ambient wind direction is a limitation of the model because its effect on wake mixing can be modeled by either changing the eddy-viscosity or by running several wind direction cases and average them using a Gaussian filter, see for example van der Laan et al. (2015a), wich is based on the work of Gaumond et al. (2014).

16. The conclusions are not based on the results of the present article:

   - *The characteristics of this model show the desired performance balance between speed and realistically achievable accuracy.* The accuracy of the model is not shown because you lack a validation.
   - *The model has been validated against performance data from offshore and onshore wind farms.* This is not performed in the present article.

**Minor comments**

1. Page 2, Line 45. You write here: *Models of this group are, in principle, also capable of solving the upstream effects of wind turbines.* You are right about the upstream effects, however, it is also the interaction of the wind turbine wakes and wind turbine induction zones, which represents the elliptic nature of these models.

2. My last name is miss-spelled in the corresponding reference (Laan should be van der Laan).

3. Page 1, Line 25: There is a typo in a citation: citetSchlez2009.

4. Section 1: You mention parabolic and elliptic solvers. While I am aware of the meaning of these terms, it would be useful to explain them in order to reach a broader audience. For example, you could mention that a parabolic solver does not need to iterate numerically and information of the flow is only transported with the flow direction, while elliptic solvers have to iterate to solve the equations and information is transported in all directions.

5. Eq. (1): There is an additional plus sign that can be removed.

**References**

Gaumond, M., Réthoré, P.-E., Ott, S., Peña, A., Bechmann, A., and Hansen, K. S.: Evaluation of the wind direction uncertainty and its impact on wake modeling at the Horns Rev offshore wind farm, Wind Energy, 17, 1169, 2014.

Iungo, G. V., Santhanagopalan, V., Ciri, U., Viola, F., Zhan, L., Rotea, M. A., and Leonardi, S.: Parabolic RANS solver for low-computational-cost simulations of wind turbine wakes, Wind Energy, 21, 184–197, https://doi.org/10.1002/we.2154, https://onlinelibrary.wiley.com/doi/abs/10.1002/we.2154, 2018.

van der Laan, M. P. and Andersen, S. J.: The turbulence scales of a wind turbine wake: A revisit of extended k-epsilon models, Journal of Physics: Conference Series, 1037, 1, https://doi.org/10.1088/1742-6596/1037/7/072001, 2018.

van der Laan, M. P., Sørensen, N. N., Réthoré, P.-E., Mann, J., Kelly, M. C., Troldborg, N., Hansen, K. S., and Murcia, J. P.: The $k\text{-}\varepsilon\text{-}f_P$ model applied to wind farms, Wind Energy, 18, 2065, https://doi.org/10.1002/we.1804, 2015a.

van der Laan, M. P., Sørensen, N. N., Réthoré, P.-E., Mann, J., Kelly, M. C., Troldborg, N., Schepers, J. G., and Machefaux, E.: An improved $k\text{-}\varepsilon$ model applied to a wind turbine wake in atmospheric turbulence, Wind Energy, 18, 889, https://doi.org/10.1002/we.1736, 2015b.

---

## Author Comment (AC1) · 31 Mar 2020

Thank you for your interest in the WakeBlaster model.

The model has been validated against data from over 20 wind farms. Some of the validations are available here: https://proplanen.info/wakeblaster, and more will follow in due course. Including the validations in the paper would have made it too long. The paper focusses on providing a clear and concise presentation of the theoretical background for the model, and the initial verification of its performance.

In a different session at the WESC, we presented validation results from one of the wind farms (Verification and Validation of the Waked Flow of a Large Wind Farm), but we did not submit a paper for review. We suggest that the editor considers if there is

any added value in including this presentation as supplementary material.

---

## Referee Comment (RC2) · Anonymous Referee #2 · 3 Apr 2020

Title:

"Verification" can be misleading since the paper does not contain any comparison with experimental data. Consider adding a plot including the validation in Lillgrund for a flow case in the results chapter. In that case, consider change 'Verification' by 'validation'

Abstract – line 10: add reference to any report including validation

Page 1 – line 16: describe further 'single turbine wake model', what implies

Page 2 – line 37: this sentence is hard to understand. Better use: *"… and later Crespo et. al. (1994) developed an extension for wind farms called UPMPARK"*

Page 2 – line 54: change 'verification' by validation. Verification implies ensures that a model is working properly (equations solved as expected, no bugs), validation implies agreement with experimental data (reality physics)

Page 2 – line 57 – suggestion: Change title of chapter 2 by "Theoretical background"?

Page 3 – line 59: unsteady terms or fluctuation term of velocity vector?

Page 3 – line 63: Avoid "We", use instead: *"Cartesian 3D vectors are used for displacement…"*

Page 3 – line 65: use streamwise and transversal components, instead of mean and lateral

Page 3 – line 70: add mass conservation equation as well at this point

Page 3 – line 74: at the pressure assumption, needs "=0" at the end

Page 4 – avoid mass conservation equation here if listed in 2.1

Page 4 – line 89 – add a new chapter here on Grid resolution and boundary conditions, there are no references except in chapter 3.1 which can be here. Also justify here why a rotor disk is composed by 80 cells as specified in the abstract, should not this value depend on rotor area?

Page 4 – line 101: specify the distance at which the near wake is placed (where the momentum deficit is injected)

Page 4 – line 103: using alfa for turbulence intensity can be misleading (same sign to refer to shear). To avoid mistakes, use TI instead

Page 5 – line 117: use transversal instead of lateral

Page 5 – line 128: remove 'a'

Page 6: general comment to chapter 2.4: since turbulence viscosity estimation (and consequently WakeBlaster) depends on those 5 parameters, a general recommendation or comment should be included (if feasible) about their range values, do they depend on the wind farm layout or scenario? (very close turbines, interaction of wakes with ground, etc.)

Page 6 – line 158: "scalar velocity" to be changed by "turbulence viscosity"

Page 6 – line 162: "…is solely based *on*.."

Page 8 – line 199: this sentence is hard to understand, please re-write

Page 9 – line 209: Chapter 3 should be dedicated to apply WakeBlaster to a particular flow case in a particular wind farm (Lillgrund). Please include a first sub section on describing Lillgrund wind farm in detail (layout, rotor diameter, etc.) and also include another section (if data were available) comparing wakeblaster and experimental efficiency values in a particular flow case.

Page 10 – line 211: Chapter on computational performance should be included on the numerical solution. This is not something inherent to the Lillgrund simulation

Page 10 – line 231: specify if the case corresponds to neutral atmosphere

Page 12 – line 250: make some reference to limitations on complex terrain. Additionally, it could be mentioned the possibility to include RSF or WRG files in order to take into account the effect of orography on the free stream flow

---

## Author Comment (AC2) · 15 Apr 2020

On model validation we would like to refer you to the previous comment in the interactive discussion(AC2). Your detailed suggestions and corrections are of great value to us and will be considered in the final paper. Thank you for your review.

---

## Author Response (AR1)

Please find below our review notes

| Page/Line | Reviewer (Paul van der Laan) Comment | Changes Made |
|---|---|---|
| Title and General | Interesting Article but a model verification and validation is missing. | The model has been comprehensively validated against data from over 20 wind farms. Some of the validations are available here: https://proplanen.info/wakeblaster, and more will following in due course. Including the validations in the paper would have made it too long, because we would have needed to include methodology etc. as well as the results. The paper focuses on providing a clear and concise presentation of the theoretical background for the model, and the initial verification of its performance. A model verification is included, as indicated by the paper's title. The verification has been expanded in the paper, and references to the validation have been added (see below). |
| Page 1, abstract | 1. In the abstract you mention:The WakeBlaster model is verified, calibrated and validated using a large volume of data from multiple onshore and offshore 10 wind farms. I cannot find a reference to this work and it is also not included in the present work. (...) In addition, please note that the abstract should include a motivation, a short summary of the work, and the main results and conclusions,so it cannot contain conclusions based on previous work. | In a different session at the WESC, the authors presented validation results from one of the wind farms (Verification and Validation of the Waked Flow of a Large Wind Farm), but a paper was not submitted. The presentation has now been uploaded to the WESC recommended repository, and a reference has been included in the paper. Additional references to further validation work and an external validation (blind test on 5 offshore wind farms) are now included. 1.1 Page, Line 55. References made in the text to validation were inconsistent with the paper's title, content and the primary purpose of the paper and so these have been removed. |
| Limitations | 2. How are the results post processed when the annual energy production is evaluated? For example, do you include a Gaussian filter to represent wind direction uncertainty? (As introduced by Gaumond et al. (2014) and applied for RANS in van der Laan et al. (2015a)). | The model presented does not include annual energy yield calculations; it is only a flow case calculator. The integration of flow cases to calculate the energy yield is done on the client-side, and therefore outside the scope of this paper. When using the model in validation studies we have (where appropriate) implemented a Gaussian filter to account for wind direction uncertainty, but this is also outside of the scope of this paper. |
| Competing Interests | 3. Since the authors are both employees at ProPlanEn, a commercial entity that is selling the presented model, it would make sense to mention this in the Section Competing interests. | The affiliation is provided in the list of authors and the funding details are provided in the acknowledgements. However, given that the reviewer comments on this, which in itself can be taken as evidence, that a potential conflict of interest could be perceived by others. It is agreed that any such perception is to be avoided. The relationships of the authors to ProPlanEn Ltd were therefore added to the competing interest section. |
| Limitations | 4. Can WakeBlaster handle (complex) terrain? If not, I would mention that the model can only used for wind farms in flat terrain and offshore conditions. | WakeBlaster relies on the underlying flow model, and makes use of the speed-ups induced by terrain and/or roughness. This is equivalent to current industry models, but it does not go beyond that. This has been added to the limitations section, as suggested. |
| Pages 2-3, Lines 58-60: | 5. I do not understand what you mean by In order to account for the unsteady terms, it uses eddy viscosity turbulence closure, where the eddy viscosity is calculated from the combined wake and ambient wind speed shear profiles. I guess you mean turbulent fluctuations instead of unsteady terms. Unsteady terms can be handled by including a transient term, as can be done in Unsteady RANS (URANS). | It is agreed that the sentence was confusing; this has been corrected. |
| | 6. The reference to Abramovich (1963) is not very accessible. In addition, eq. (2) of the article is the boundary layer equation without viscous effects and I do not understand how this equation can be extended to three dimensions because the original boundary layer equation describes a streamwise $U$ and vertical velocity, which in your coordinate system is $W$ not $V$ . I would suggest to start Section 2.1 with the 3D RANS equations including external forces ... | Abramovich 1963 (MIT Press) is a classical textbook on the topic of turbulent jets, which is still in print and available from any university library by inter-library loan, or from Amazon. It is geared to experiments and engineering applications, and is well known. It is considered preferable to to provide a reference to reference to providing a lengthy derivation from first principles. |
| | 6. (...) in order to arrive at eq. (5) of the article we need to assume that $\partial V/\partial x$ and $\partial W/\partial x$ are zero, which is not mentioned in the article. The assumptions (a) and (c) are neither mentioned. Assumptions: a) The momentum equations for v and w are ignored. d) The gradient of the normal Reynolds stress in the streamwise direction is zero. | The assumptions have been revised. Changes have also been made to section 3.1, to clarify that we are looking at a free jet and modelling the momentum in the flow direction. |
| | 6. (...) We assume the eddy viscosity to be variable. | The eddy viscosity changes downstream, throughout the domain, as the wake decays. It is assumed that the eddy viscosity is constant across the wake. No attempt is made to model the fine structure in the near wake. |
| | 7. You mention that the near wake stream-wise velocity profile is prescribed for each wind turbine. How do you determine the end of the near wake and how does it vary with the wind turbine thrust coefficient and atmospheric conditions as turbulence intensity and stability? Do I understand correctly that eq. (7) is used to determine the initial magnitude of the centreline wake deficit at a defined downstream location? In addition, eq. (7) is derived from wind tunnel measurements where the turbulence length scales and Reynolds-number are very different from utility scale turbines, so would that mean eq. (7) needs to be recalibrated? | Yes, the fundamental work by Ainslie is used for the initial centerline wake deficit. Wq. (7) defines the wake deficit at a fixed downstream distance of 2D. This approximation has proven reliable over decades of use in various engineering codes. It is derived from a mix of field experiments (MW turbines) and wind tunnel experiments. Some of the data used to derive (7) is problematic and a re-calibration has been discussed. However given that the application of this equation has a long history in the industry. and that there is an absence of sufficient full-scale data to re-calibrate the function, use of the equation remains a reasonable choice. |

| | | |
|---|---|---|
| | 7. (...) Finally, I was wondering why you are not modeling the wind turbine thrust force as an external force in the streamwise momentum equation instead of prescribing a velocity deficit profile in the near wake. | For a detailed model of the near wake, one would need to set the radial distribution of the axial force, and use a model for the extraction of momentum in the induction zone and near wake. This requires the flow expansion to be modelled, including the radial and tangential flow components and the elliptic effects. The model would increase in complexity, beyond what can be validated. It would also become computationally which is unnecessary for practical application. The approach introduced by Lissaman and Ainslie is ingenious, in that it avoids (with acceptable loss of accuracy) the complexity of modelling the near wake. Ongoing efforts to re-calibrate the function (7) are, yet, inconclusive. |
| Section 2.4 | 8. The eddy-viscosity is no longer a constant, as assumed in eq. (5) of the article, which is inconsistent. Please motivate and clarify. | The eddy viscosity changes are at a slow time/length scale, over the domain and while the wake develops. In the near wake there are strong radial changes and blade tip vortices. Unlike in the wind tunnel, in a field experiment these small scale turbulence decays quickly. Outside of the near wake, the eddy viscosity changes more slowly. Turbulent eddies have the scale of the rotor diameter and can be approximated as constant over the length scale. Over longer time/length scale eddy-viscosity is not constant, as it is a property of the flow rather than the fluid, and it changes in all directions over the domain. We have added a sentence in section 2.1, to clarify this approximation. |
| Page 6 Line 158 | 9. There are a number of undefined parameters and constants. What are the values of ??? (Page 6, line 158), ??? (eq. 13), ??? (eq. 14), ??? (eq. 14)? | All parameters are now defined. |
| | 10. Should k be $\kappa$ in eq. (14)? | The naming of k is correct. It refers to the eddy viscosity calibration constant (section 2.4, bullet point 4). However, I do note that it could be easily redefined by substituting $\phi' = \frac{\phi}{k}$. |
| Section 2.4.1 | 11. . I suspect that the eddy-viscosity lag model could be replaced or simplified using a length scale limiter in the form of an fP function (...) that only has one constant to calibrate, see for example van der Laan et al. (2015b) or van der Laan and Andersen (2018) | We agree that the fp approach is similar and that it could be useful to compare them at some point in the future. The 'fixed' eddy viscosity lag model also has only one calibration parameter. |
| Section 3.1 | 12. : Please motivate the chosen grid resolution of D/10, where D represent the rotor diameter, using a grid refinement study. In addition, a domain height of 3D seems very low to me, please show that this domain height does not affect the solution. I normally use 25D for 3D elliptic RANS simulations of wind farms. What are the other dimensions of the 3D flow domain? Do you use stretching of cells in order to reduce the total number of cells or is the domain discretized uniformly? | The resolution is balanced with processing speed. It must be high enough to represent wakes (in the order of magnitude of the rotor diameter) and of sufficient detail for modelling partial wakes, wake interaction, and wake boundary layer interaction. The grid is a uniform rectangular structured grid with dx=dy=dz. The simple structure simplifies the model implementation and avoids numerical challenges at grid transitions. Additional vertical layers are not required for numerical stability, because we are using a downstream marching solution. Comparing the domain with an elliptical solver is not really valid - it is not surprising to us that an elliptical solver would need a larger domain. More information on the grid has been added in section 2.3.1. |
| Section 3.1: | 13. You mention that a flow case of Horns Rev I takes 5 s. (Please briefly introduce the Horns Rev I wind farm here). That would mean an annual energy production calculation of 22 wind speeds and 360 wind directions would take 11 hours on a single CPU. This can be made parallel as you mention (using a few hundred cores). However, WakeBlaster should provide more accurate results compared to an engineering wake model that can calculate the AEP in about 1 s on a single core in order to make sense to run. Therefore, I would suggest to both validate WakeBlaster with wind farm measurements and compare the performance with one or two engineering wake models in the present work. | Yes, the AEP calculation consists typically of 2,500-10,000 flow cases, or 55,000 if you calculate the AEP for a representative year, at a time-step of 10 minutes. In the third party OWA blind tests, WakeBlaster delivered results with a more detailed structure and a reduced standard deviation. This blind test can also be interpreted as a practice test, where we were able to show that it is realistic to use CFD in an iterative process. The description of Horns Rev was added here, and references to both our own validations and third party offshore validation were added to section 1.1. |
| Section 3.2 | 14. You could use this wind farm as both a verification (grid study) and validation case. Presenting a show case is not enough for a scientific document. | The verification has been expanded and references to validation have been added. |
| Section 4: | 15. You could add that only flat terrain is considered. In addition, I do not agree that meandering of the ambient wind direction is a limitation of the model because its effect on wake mixing can be modeled by either changing the eddy-viscosity or by running several wind direction cases and average them using a Gaussian filter, see for example van der Laan et al. (2015a), wich is based on the work of Gaumond et al. (2014). | The limitation referring to meandering has been removed, and limitations with respect to terrain have been added. |
| Conclusion | 16. The conclusions are not based on the results of the present article: (...) | Agree, fixed (reference to validation removed) |
| Page 2, Line 45. | 1. You write here: Models of this group are, in principle, also capable of solving the upstream effects of wind turbines. You are right about the upstream effects, however, it is also the interaction of the wind turbine wakes and wind turbine induction zones, which represents the elliptic nature of these models. | The text in section 1.1 was expanded accordingly. |
| Reference | My last name is miss-spelled in the corresponding reference (Laan should be van der Laan) | Apologies, corrected. |
| Page 1, Line 25: | 3. There is a typo in a citation: citetSchlez2009. | Corrected |
| Section 1: | 4. You mention parabolic and elliptic solvers. While I am aware of the meaning of these terms, it would be useful to explain them in order to reach a broader audience. For example, you could mention that a parabolic solver does not need to iterate numerically and information of the flow is only transported with the flow direction, while elliptic solvers have to iterate to solve the equations and information is transported in all directions. | Text in section 1.1 was expanded accordingly. |
| Eq. (1): | 5. There is an additional plus sign that can be removed. | Corrected |
| | | Thank you for the review. |

Please find below our review notes

| Page/Line | Anonymous Reviewer 2 | Changes Made |
|---|---|---|
| Title | "Verification" can be misleading since the paper does not contain any comparison with experimental data. Consider adding a plot including the validation in Lillgrund for a flow case in the results chapter. In that case, consider change 'Verification' by 'validation' | "Verification" is not misleading in this instance. "Verification" is an evaluation of how a product meets its specification and expectations, which is covered in chapter 3. "Validation" would be a comparison with experimental data, and it is not presented in this paper, in order to keep it concise. |
| Abstract, Line 10 | add reference to any report including validation | References to model validations were added to section 1.1. |
| Page 1 – line 16: | describe further 'single turbine wake model', what implies | The description has been expanded. |
| Page 2 – line 37 | this sentence is hard to understand. Better use: "… and later Crespo et. al. (1994) developed an extension for wind farms called UPMPARK" | Done. |
| Page 2 – line 54 | change 'verification' by validation. Verification implies ensures that a model is working properly (equations solved as expected, no bugs), validation implies agreement with experimental data (reality physics) | Verification was intentionally chosen, as the paper in its current form does not include a validation. Inconsistent references to validation have been removed throughout the text. |
| Page 2 – line 57 | suggestion: Change title of chapter 2 by "Theoretical background"? | Done. |
| Page 3 – line 59 | unsteady terms or fluctuation term of velocity vector? | This has been rephrased and clarified in the text. |
| Page 3 – line 63: | Avoid "We", use instead: "Cartesian 3D vectors are used for displacement…" | Changed in the four relevant instances lines (49, 69, 166, 303) in the text. |
| Page 3 – line 65: | use streamwise and transversal components, instead of mean and lateral | "Mean" and "lateral" have been replaced by "streamwise", "horizontal" and "vertical". Transversal has been used to address both horizontal and vertical directions. |
| Page 3 – line 70: | add mass conservation equation as well at this point | Done. |
| Page 3 – line 74: | at the pressure assumption, needs "=0" at the end | Done. |
| Page 4 | avoid mass conservation equation here if listed in 2.1 | Done. |
| Page 4 – line 89 | add a new chapter here on Grid resolution and boundary conditions, there are no references except in chapter 3.1 which can be here. (…) | Added to 2.3. |
| Page 4 – line 89 | (…) Also justify here why a rotor disk is composed by 80 cells as specified in the abstract, should not this value depend on rotor area? | The high resolution is required to capture the wake profile for full and partial wakes with sufficient accuracy. The default model resolution is 0.1 turbine diameters, which makes it independent of the rotor area. At this resolution the rotor is covered by an average of $10*10*\frac{\pi}{4} / \approx 80$ points. However, this number varies from rotor to rotor, as the grid is not snapped to any individual turbine rotor. The number has been deleted in the revised text. |
| Page 4 – line 101: | specify the distance at which the near wake is placed (where the momentum deficit is injected) | Done. |
| Page 4 – line 103: | using alfa for turbulence intensity can be misleading (same sign to refer to shear). To avoid mistakes, use TI instead | Done - now using I for turbulence intensity, and the left hand side of the equation has been corrected. |
| Page 5 – line 117 | use transversal instead of lateral | Done. |
| Page 5 – line 128 | remove 'a' | Done. |
| Page 6 ch 2.4 | general comment to chapter 2.4: since turbulence viscosity estimation (and consequently WakeBlaster) depends on those 5 parameters, a general recommendation or comment should be included (if feasible) about their range values, do they depend on the wind farm layout or scenario? (very close turbines, interaction of wakes with ground, etc.) | The description of the parameters has been expanded. They are independent of the wind farm layout and scenario. |
| Page 6 – line 158: | scalar velocity" to be changed by "turbulence viscosity" | This sentence has been corrected and rephrased. |
| Page 6 – line 162: | "… is solely based on.." | Done. |
| Page 8 – line 199: | this sentence is hard to understand, please re-write | Sentence was rewritten. |
| Page 9 – line 209 | Chapter 3 should be dedicated to apply WakeBlaster to a particular flow case in a particular wind farm (Lillgrund). Please include a first sub section on describing Lillgrund wind farm in detail (layout, rotor diameter, etc.) and also include another section (if data were available) comparing wakeblaster and experimental efficiency values in a particular flow case. | Lillgrund wind farm details added and renamed Verification for consistency throughout the text. |
| Page 10 – line 211 | Chapter on computational performance should be included on the numerical solution. This is not something inherent to the Lillgrund simulation | Chapter 3 has been restructured into two sections. Section 3.1 covers computational performance verification, and Section 2.3 is dedicated to implementation of the numerical solution. |
| Page 10 – line 231: | specify if the case corresponds to neutral atmosphere | Done. |
| Page 12 – line 250: | make some reference to limitations on complex terrain. Additionally, it could be mentioned the possibility to include RSF or WRG files in order to take into account the effect of orography on the free stream flow | Done. |
|  |  | Thank you for the review. |

[revised manuscript text omitted]

---

## Referee Report (RR1)

**Review of *Theory and Verification of a new 3D RANS Wake Model (revised version R1), by Philip Bradstock and Wolfgang Schlez**

Reviewer: M. Paul van der Laan, DTU Wind Energy

Overall, the authors have responded correctly to most of my suggestions and comments. However, I do have new concerns regarding the grid refinement study that has been added. Since the article is called ...*Verification of a new 3D RANS Wake Model*, then model verification in the form of a grid refinement study should addressed sufficiently in order to accept the article for publication.

**Main comments**

1. It is great that you have added a grid refinement study (Section 3.1), although the results presented in the Figure 2 indicate that either the RANS model does not behave well or something in the grid refinement study is not set up correctly. The numerical discretization error of a well behaving RANS model should decrease monotonically with grid refinement, which is minimal requirement. For example, if the order of the employed numerical scheme(s) is 2, then one would expect to obtain a reduced discretization error by a factor of 4 when the grid is refined by a factor of 2. Figure 2 shows that the absolute value of the chosen error metric increases with grid refinement, which is not acceptable for a RANS model. In other words, if one would choose to apply the RANS model with a finer resolution, then one would end up with a larger discretization error. As a result, a conclusion based on a simulation with the chosen grid size might not be valid for a simulation with a much finer grid. Figure 2 could indicate that the model needs a much finer resolution in order to provide well behaved discretization error.

   Furthermore, it is misleading to normalize the error by the error from your chosen grid size. Instead, it is better to normalize the error by a Richardson extrapolated value, which would represent the solution for a zero discretization error. If you have employed numerical schemes with different discretization orders you could apply a mixed order analysis as suggested by Roy (2003) and applied to a grid refinement study of single wake RANS simulations in Réthoré et al. (2013) and van der Laan et al. (2015). This method also reveals the leading order of the discretization error. However, if the discretization error of the RANS models does not monotonically decrease with grid refinement, then the mixed order analysis cannot be used. In this case, one should normalized by the error/ wind speed of the finest grid and conclude that the model is grid dependent.

   In addition, the information provided in Section 3.1 is not sufficient to understand and redo the grid refinement study: I lack information on:

   (a) How is the error defined? Where is the wind speed extracted in the lateral and vertical (height) dimensions? (You only mention the downstream distance.) Does the error of wind speed represent a point value or is it an integral over a chosen area (e.g. the rotor area representing a virtual downstream wind turbine)? Using an integral value is often a better metric for a grid refinement study.

   (b) What are the values of the thrust coefficients for both wind turbines and what is the wind turbine type that you used?

   (c) What is the distance between the two wind turbines and what is the wind direction relative to the wind farm layout?

2. It is good that you have moved the validation from the conclusion but you also need to remove or rephrase/explain the word accuracy in the conclusion because you have not tested the model accuracy using a validation study.

**References**

Réthoré, P.-E., van der Laan, M. P., Troldborg, N., Zahle, F., and Sørensen, N. N.: Verification and validation of an actuator disc model, Wind Energy, pub. online, 2013.

Roy, C. J.: Grid Convergence Error Analysis for Mixed-Order Numerical Schemes, American Institute of Aeronautics and Astronautics Journal, 41, 2003.

van der Laan, M. P., Sørensen, N. N., Réthoré, P.-E., Mann, J., Kelly, M. C., Troldborg, N., Schepers, J. G., and Machefaux, E.: An improved $k$-$\varepsilon$ model applied to a wind turbine wake in atmospheric turbulence, Wind Energy, 18, 889, https://doi.org/10.1002/we.1736, 2015.

---

## Author Response (AR2)

Please find below our review notes

| Page/Line | Reviewer (Paul van der Laan) Comment | Changes Made |
|---|---|---|
| Section 3.1, numbers 230-260 | Figure 2 indicate that either the RANS model does not behave well or something in the grid refinement study is not set up correctly. Furthermore, it is misleading to normalize the error by the error from your chosen grid size. Instead, it is better to normalize the error by a Richardson extrapolated value, which would represent the solution for a zero discretization error. | We have redone the grid refinement study in section 3.1, improved the description of the grid refinement study in section 3.1, and normalised as requested in figure 2 the error by a Richardson extrapolated value for zero discretisation. |
| Section 3.1, number 245 | ... the information provided in Section 3.1 is not sufficient to understand and redo the grid refinement study: I lack information (...) (a) How is the error defined? Where is the wind speed extracted in the lateral and vertical (height) dimensions? (You only mention the downstream distance.) Does the error of wind speed represent a point value or is it an integral over a chosen area (e.g. the rotor area representing a virtual downstream wind turbine)? Using an integral value is often a better metric for a grid refinement study. | The error has been updated and is not defined relative to the Richardson extrapolated value for zero discretisation. The setup has been clarified and the requested detail has been added. |
| Section 3.1, number 245-250 | ... the information provided in Section 3.1 is not sufficient to understand and redo the grid refinement study: I lack information (...) (b) What are the values of the thrust coefficients for both wind turbines and what is the wind turbine type that you used? (c) What is the distance between the two wind turbines and what is the wind direction relative to the wind farm layout? | Apologies, the description of the setup was misleading, now clarified and information on the turbine (type, thrust coefficient) added. |
| Section 5., number 305 | It is good that you have moved the validation from the conclusion but you also need to remove or rephrase/explain the word accuracy in the conclusion because you have not tested the model accuracy using a validation study. | We have modified the conclusions accordingly. |
| Section 2.4, number 170 | | Spelling mistake corrected |
| | | Thank you for the review. |

[revised manuscript text omitted]